# Neurosymbolic Sentiment Analysis
# with Dynamic Word Sense Disambiguation

**Xulang Zhang♠, Rui Mao♠, Kai He♣ and Erik Cambria♠**
♠Nanyang Technological University, Singapore
♣National University of Singapore, Singapore
xulang001@e.ntu.edu.sg; {rui.mao,cambria}@ntu.edu.sg; kai_he@nus.edu.sg

## Abstract

Sentiment analysis is a task that highly depends on the understanding of word senses. Traditional neural network models are black boxes that represent word senses as vectors that are uninterpretable for humans. On the other hand, the application of Word Sense Disambiguation (WSD) systems in downstream tasks poses challenges regarding i) which words need to be disambiguated, and ii) how to model explicit word senses into easily understandable terms for a downstream model. This work proposes a neurosymbolic framework that incorporates WSD by identifying and paraphrasing ambiguous words to improve the accuracy of sentiment predictions. The framework allows us to understand which words are paraphrased into which semantically unequivocal words, thus enabling a downstream task model to gain both accuracy and interpretability. To better fine-tune a lexical substitution model for WSD on a downstream task without ground-truth word sense labels, we leverage dynamic rewarding to jointly train sentiment analysis and lexical substitution models. Our framework proves to effectively improve the performance of sentiment analysis on corpora from different domains.

## 1 Introduction

To improve the accuracy of sentiment analysis, it is crucial to disambiguate polysemous words according to the context (Cambria et al., 2023). A word may have senses that carry opposite sentiments, e.g., *"fine"* has the positive meaning of being delicate and refined, and the negative meaning of monetary penalty. Additionally, words may appear in positive, negative, or neutral senses in different domains. For instance, the word *"bearish"* is often used in a negative sense to describe the stock market in the financial domain, while more likely to be used neutrally to describe hulking figures in other domains.

In traditional black-box sentiment analysis, the representations of word meanings are generated by neural networks (Mao and Li, 2021). As such, the disambiguation process is implicit and opaque. It is challenging to investigate what implicit senses are assigned to ambiguous words by a neural network model for it to make accurate predictions. Coupling symbolism and sub-symbolism, a neurosymbolic approach to sentiment analysis is transparent and interpretable, regarding the extraction and distinction of word meaning as a sub-problem (Cambria et al., 2017). However, research that explicitly deals with ambiguous words in sentiment analysis is limited (Xia et al., 2014, 2015; Cambria et al., 2015). Sense-annotated resources such as SentiWordNet (Baccianella et al., 2010) have been utilized, but only to detect and score sentimental words in the input (Hung and Lin, 2013; Nassir-toussi et al., 2015; Hung and Chen, 2016). The disambiguation of polysemous words in sentiment analysis inputs remains under-explored. Thus, we are motivated to develop a neurosymbolic framework for sentiment analysis with explicit Word Sense Disambiguation (WSD).

Traditional WSD aims to classify which sense of the target word is appropriate according to its context (Huang et al., 2019; Blevins and Zettlemoyer, 2020; Bevilacqua and Navigli, 2020; Song et al., 2021; Barba et al., 2021a,b). However, the limitations ($\ell$) of their task setups are that ($\ell1$) the output of those WSD systems, whether sense label (Bevilacqua and Navigli, 2020) or gloss definition (Blevins and Zettlemoyer, 2020; Barba et al., 2021a), is difficult to integrate into downstream input; ($\ell2$) the target words must be pre-defined for WSD systems to disambiguate, making it difficult to apply them directly to downstream tasks; and ($\ell3$) the reliability of sense distinction of the benchmark has been questioned (Ramsey, 2017; Maru et al., 2022), as word senses are not discrete.

Attempts have been made to incorporate the disambiguation of word senses into downstream applications. Nonetheless, they are either impractical in real-world settings (Farooq et al., 2015; Pu et al., 2018), or only applicable to a specific downstream task, i.e., machine translation (Hangya et al., 2021; Campolungo et al., 2022). To address these issues, we leverage dynamic WSD in our neurosymbolic framework. We regard WSD as a lexical substitution task (for $\ell 1$), where ambiguous words in sentiment analysis input are paraphrased into ones that cause less sentiment confusion using a lexical substitution model with dynamic WSD.

Our dynamic WSD method comprises two distinct aspects: the dynamic selection of the target word for disambiguation, based on downstream contextual information (for $\ell 2$); and the dynamic learning of the most suitable lexical substitutions, based on the prediction accuracy of the downstream model (for $\ell 3$). To identify target words that need to be substituted, we first adopt an attention-based explainable encoder to rank input words by the level of influence they have on the output, motivated by the assumption that not all polysemous words affect sentiment predictions (Cao et al., 2015). Then, the top-ranking words with the most diverse senses are selected to be targets, based on our hypothesis that an ambiguous word may coincide with significant disparities in the semantic space between the word itself and its potential substitution candidates.

As such, we can trace back which words are selected as targets, and which substitute words are used to disambiguate them, making our method interpretable and transparent. Furthermore, we fine-tune the lexical substitution model whilst training the sentiment analysis model by adopting a dynamic rewarding mechanism that does not require gold standard substitution labels. As such, the substitution model can be adjusted to provide more appropriate output for the downstream task as well as the target domain. Our model achieves more accurate sentiment analysis with explicit lexical-substitution-based WSD, despite not specifically trained on WSD and sentiment parallel datasets. Since the disambiguated word senses are represented as the lexical substitutions of the original words and an explainable encoder is used for retrieving the most contributing words, the neurosymbolic sentiment inference process is more interpretable for humans compared to black-box models.

Our model is examined with three publicly available datasets from movie review, finance, and Twitter domains, outperforming strong baselines by 1.23% on average. The contributions of our work are twofold: (1) We propose a neurosymbolic sentiment analysis framework that leverages WSD as an auxiliary lexical substitution task, achieving state-of-the-art performance on our examined sentiment analysis datasets from different domains; (2) We design a transparent mechanism to select WSD target words, based on a) the importance of words to the learning and inference of a downstream task, and b) word sense diversity.

## 2 Related Work

**Word Sense Disambiguation** is a task that assigns the appropriate meaning to an ambiguous word according to its context. Recent WSD systems attained F1 scores higher than 80% on the benchmark datasets (Song et al., 2021; Bevilacqua and Navigli, 2020; Barba et al., 2021a,b), which is regarded as the estimated human performance since it is the highest inter-annotator agreement (Edmonds and Kilgarriff, 2002; Palmer et al., 2007). However, research on the application of WSD systems on downstream tasks is limited. Pu et al. (2018) utilized clustering algorithms to obtain sense embeddings by inferring from the similarity of the context vectors of a target word. The heavy requirement for disambiguating every homographs prior to downstream applications makes their method less feasible in many real-world situations. Campolungo et al. (2022) achieved the disambiguation of word sense in neural machine translation by leveraging a multilingual WSD system to establish a sense-tagged parallel corpus. Nonetheless, their approach relied on cross-lingual word alignments for reliable annotations, meaning that it cannot be adopted by other types of downstream tasks.

**Sentiment Analysis** can be generally defined as the task that determines if a piece of text expresses positive, negative, or neutral opinion (Mao et al., 2023c). Word sense information has been applied in sentiment analysis in the form of sentiment-annotated lexicons (Devitt and Ahmad, 2007; Ohana and Tierney, 2009; Hung and Lin, 2013; Nassirtoussi et al., 2015), such as SentiWordNet. However, these methods incorporate the sentiment information of a word without disambiguating its word sense according to the context, but resorting to the emotional score of the most frequent

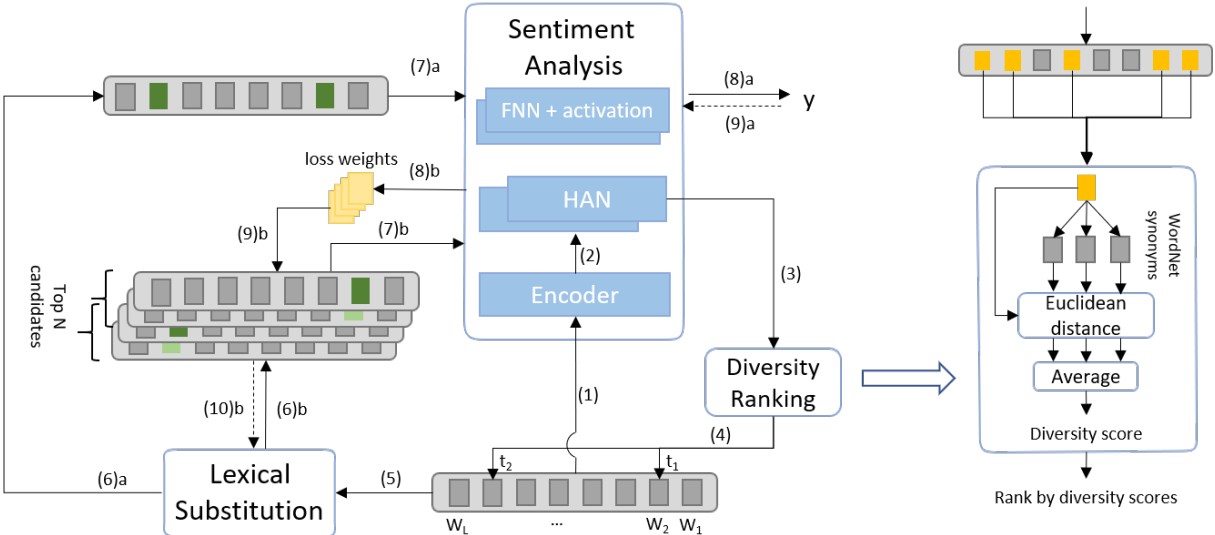

Figure 1: Architecture of the proposed disambiguation framework for sentiment analysis. A dotted line means backpropagation. Green block means that the original word is replaced by a word provided by the lexical substitution model. The darker the green, the higher probability it is assigned to by the model. Orange block indicates the words with top ranking attention scores produced by HAN.

sense, or the average score of all senses. Additionally, the main objective of this approach is to identify sentimental words in a sentence, instead of disambiguating polysemous words that might affect model predictions. Hung and Chen (2016); Farooq et al. (2015); Liu et al. (2018) attempted to disambiguate according to target domain and context, but either costed extensive human labour, or fail to incorporate word senses explicitly and transparently.

To address these limitations, our neurosymbolic framework aims to identify ambiguous target words in the context and replace them with clarifying ones automatically, making them interpretable. We also leverage a dynamic rewarding mechanism so that the substitutions can be fine-tuned to better serve the downstream task and the text domain.

## 3 Task Definition

In this work, we regard word sense ambiguity as the potential cause of errors in a downstream task, e.g., sentiment analysis. Inspired by the work of Mao et al. (2023b), we reframe WSD as a lexical substitution task, aiming to replace an ambiguous word with a synonym that not only represents the true sense of the original word in the context (Arthur et al., 2015), but also reduces misclassifications in the downstream task. Given an input sentence $w = (w_1, w_2, \ldots, w_L)$, we first aim to identify the target word $t$ that is likely troublesome for

the downstream prediction from $w$. To resolve the potential ambiguity, we extract $M$ of its synonyms from WordNet (Fellbaum, 1998) as candidates $c = (c_1, \ldots, c_M)$, forming the input $(s)$ as

$$s = \text{}, w_1, w_2, \ldots, t, \ldots, w_L, \text{},$$
$$c_1, \text{}, c_2, \text{}, \ldots, c_M, \text{}. \quad (1)$$

 and  are special tokens that were defined by our employed pre-trained language model. The lexical substitution model should identify the best candidate from $c$ as the substitute $\hat{c}$ which retains the original meaning of $t$ in the context by using contrast learning. Thus, $\hat{c}$ is the symbolic representation of $t$ in context $w$. Then, the paraphrased input $w^f = (w_1, \ldots, \hat{c}, \ldots, w_L)$ is fed into a neural network classifier to predict a sentiment label. The objective is that the paraphrased input $w^f$ increases the probability of correct sentiment prediction.

## 4 Methodology

Our proposed neurosymbolic framework can be viewed in Figure 1. There are two models in our framework, namely sentiment analysis and lexical substitution models. The sentiment analysis model consists of an encoder, an interpretable attention module, and multiple feedforward neural networks. First, through Figure 1 (1) and (2), the original input is fed into the encoder and the interpretable attention module to obtain the top $I$ tokens with the highest attention weights, which contribute the most to sentiment prediction.

In (3), the sense diversity of each token is computed as the average distance of the token's hidden state to those of its substitution candidates. Then, through (4), top $J$ tokens with the most diverse senses are selected as disambiguation targets, because these target tokens may have very different senses in different contexts. Subsequently, the pre-trained lexical substitution model is applied to provide the best substitution to replace each target token, as shown in (5) and (6)a. The new input sentence is passed onto the sentiment analysis model for final prediction ((7)a and (8)a), which is used for the sentiment model backpropagation ((9)a).

In order to fine-tune the lexical substitution model to provide better substitutions that improve the accuracy of sentiment analysis, we adapt a dynamic rewarding mechanism (Ge et al., 2022) that was proposed for multi-task learning. As shown in Figure 1 (5) and (6)b, for each target word, we choose the top $N$ candidates. Each of them is seen as a substitute candidate to calculate the probability of correct sentiment prediction after the replacement ((7)b and (8)b). Then, each probability is used to dynamically compute the loss weight when the corresponding candidate is learned by the lexical substitution model as a ground truth ((9)b and (10)b). If the replacement by a candidate leads to a higher chance of predicting the right sentiment, the candidate is more likely given a higher rank by the lexical substitution model. Otherwise, the model would be less likely to select the candidate as the best replacement. The important notations used in this paper are summarized in Table 1. The detailed training process can be seen in Algorithm 1.

## 4.1 Sentiment Analysis with Interpretability

Given input sentence $w = (w_1, \ldots, w_L)$, the goal is to predict the correct sentiment label $\tilde{y}$. The input is first fed into a pre-trained encoder:

$$V = Encoder(w), \qquad (2)$$

where $V$ is hidden states. Next, we aim to find the tokens that contribute the most to sentiment inference. We adopt an interpretable attention module called Hierarchical Attention Network (HAN, Han et al., 2022), which effectively encodes hidden states with multiple non-linear projections and ranks the most influential tokens based on attention. Following the setup of Han et al. (2022), we stack two blocks of HAN to form our attention module.

$$q, a = HAN(HAN(V)), \qquad (3)$$

Table 1: Notation table.

| | Description |
|---|---|
| $L$ | Input sequence length |
| $I$ | Number of words with top ranking attention scores to be potential target words |
| $J$ | Number of words with top ranking diversity scores to be potential target words |
| $N$ | Number of candidates used for finetuning lexical substitution via dynamic rewarding |
| $K$ | Number of candidates for training lexical substitution model |
| $w$ | The original input sentence |
| $V$ | The hidden states of $w$ |
| $\tilde{y}$ | The ground-truth sentiment label of $w$ |
| $w^f$ | The final input sentence where ambiguous words in $w$ are substituted |
| $\hat{y}^f$ | The predicted sentiment label with $w^f$ as input |
| $w_l$ | The $l$-th word in $w$ |
| $a_l$ | The attention weight of the $l$-th word in $w$ |
| $w_i^{att}$ | The word in $w$ with the $i$-th highest attention weight |
| $v_i^{att}$ | The hidden state of $w_i^{att}$ |
| $M$ | The number of synonyms of $w_i^{att}$ with the same part-of-speech that are provided by the WordNet |
| $G$ | The hidden states of the $M$ WordNet synonyms |
| $g_m$ | The hidden state of the $m$-th WordNet synonyms of $w_i^{att}$ |
| $d_i$ | The average Euclidean distance between $v_i^{att}$ and $g_1, \ldots, g_m, \ldots, g_M$ |
| $t_j$ | The $j$-th target word in $w$ to be substituted |
| $c$ | A set of candidate words from WordNet. |
| $\hat{c}$ | The top $N$ candidates ranked by the lexical substitution model |
| $\hat{c}_{j,n}$ | The top $n$-th candidate for replacing $t_j$ |
| $w^{j,n}$ | The resulting sentence when replacing $t_j$ in $w$ with $\hat{c}_{j,n}$ |
| $\hat{\theta}_{j,n}$ | The loss weight when $\hat{c}_{j,n}$ is treated as the gold standard substitution |
| $s$ | The formatted input to lexical substitution model |
| $U$ | The hidden state matrix of the input sentence in $s$ |
| $u_l$ | The hidden state of the $l$-th input word in $s$ |
| $R$ | The hidden state matrix of the candidates in $s$ |
| $r_k$ | The hidden state of the $k$-th candidate in $s$ |
| $\tau$ | The temperature hyper-parameter for training the lexical substitution model |
| $\beta$ | The hyper-parameter for balancing the sentiment analysis loss and fine-tuning loss |
| $\mathcal{L}^{(sa)}$ | The sentiment analysis loss |
| $\mathcal{L}^{(tune)}$ | The loss for finetuning the pretrained language model into a lexical substitution model |
| $\mathcal{L}_{j,n}^{(ls)}$ | The lexical substitution loss when $\hat{c}_{j,n}$ is treated as the gold standard substitution |
| $\mathcal{L}^{(ls)}$ | The final weighted lexical substitution loss for backpropagation |

where vector $q$ is the yielded hidden state. $a$ is the attention weights, indicating the contribution of each token to the final sentiment prediction. To obtain the sentiment prediction, $q$ is passed on to two layers of feed-forward neural networks (FNN) to obtain the probability distribution of sentiment prediction, with the first one being activated by ReLU (Agarap, 2018), and the second by softmax.

$$h = ReLU(FNN_1(q)) \qquad (4)$$

$$\hat{y} = softmax(FNN_2(h)) \qquad (5)$$

We denote the prediction of the sentiment analysis model as $\hat{y}^f$ when the input is $w^f$, which denotes a paraphrased $w$ where all selected target tokens are replaced by sense-clarifying substitutions. Thus, the sentiment analysis loss is computed as:

$$\mathcal{L}^{(sa)} = CrossEntropy(\hat{y}^f, \tilde{y}) \qquad (6)$$

### 4.2 Disambiguation by Lexical Substitution

To obtain a lexical substitution model, we fine-tune the pre-trained language model ALM (Mao et al., 2023a)[1] with benchmark datasets from McCarthy and Navigli (2007); Kremer et al. (2014). We use the candidates from the datasets to formulate our input $s$ as in Formula 1. The candidate with the highest score is set as the ground truth substitution $\tilde{c}$. Given the input $s$, the model encodes it as:

$$U, R = ALM(s), \qquad (7)$$

where $U = [u_1, \dots, u_L]$ is the hidden states of the input sentence, and $R = [r_1, \dots, r_K]$ is the hidden states of the candidates. We denote the representation of the target word as $u_t$ ($t \in \{1, \dots, L\}$).

Our training objective is to a) close the distance between the representation of the ground truth candidate $r_k$, ($k \in \{1, \dots, K\}$) with $u_t$, and b) push away incorrect candidates representations $r_j$ ($j \in \{1, \dots, K | j \neq k\}$) from $u_t$. Namely, ($r_k, u_t$) will be regarded as a positive pair, while ($r_j, u_t$) will be regarded as a negative pair. We follow the InfoNCE loss (Oord et al., 2018) to achieve these goals, which can be formulated as :

$$\mathcal{L}^{(tune)} = -\sum_i \log \frac{exp(d(u_t, r_k)/\tau)}{\sum_j exp(d(u_t, r_i)/\tau)}, \quad (8)$$

where $i \in \{1, \dots, K\}$, $\tau$ is a temperature hyper-parameter, and $d(\cdot)$ is Euclidean distance.

---

[1]ALM employed a novel pre-training paradigm, termed Anomalous Language Modeling. ALM was pre-trained to detect anomalous substituted words from a sequence and retrieve the original words from a set of candidates that contains a positive sample (an original word) and multiple hard negative samples (the synonyms of the original word) via contrastive learning. Thus, it is more suitable for fine-tuning a lexical substitution model in our task, because we also aim to detect appropriate substitution words from a set of candidates.

**Algorithm 1:** Sentiment analysis with word sense disambiguation using a dynamic rewarding mechanism.

---

1   Initialize sentiment analysis model as $\Phi$, pre-trained lexical substitution model as $\Psi$ ;
2   Initialize hyperparameters $\beta$, $I$,$J$, $N$;
3   **while** *not done* **do**
4      Sample a sentence $w = w_1, w_2, \dots, w_L$;
5      **for** *l=1:L* **do**
6          Compute the attention weight $a_l$ of token $w_l$;
7      **end**
8      $w^{att} \leftarrow$ Top $I$ of $w$ ordered by attention weights $a = a_1, a_2, \dots, a_L$;
9      **for** *i=1:I* **do**
10         $c \leftarrow$ all possible candidates with the same part-of-speech type as $w_i^{att}$, provided by WordNet;
11         Compute the average Eulidean distance $d_i$ between the hidden states of $w_i^{att}$ and $c$, produced by the encoder in $\Phi$;
12      **end**
13      $t = (t_1, t_2, \dots, t_J) \leftarrow$ Top $J$ of $w^{att}$ ordered by $d = d_1, d_2, \dots, d_I$;
14      $w^f \leftarrow w$ ;
15      **for** *j=1:J* **do**
16         Input $w$ into $\Psi$ to produce top $N$ candidates $\hat{c}_j = (\hat{c}_{j,1}, \dots, \hat{c}_{j,N})$ for target word $t_j$, ordered by probability;
17         Replace $t_j$ in $w^f$ with $\hat{c}_{j,1}$;
18         $s \leftarrow$ (, $w_1, \dots, t_i, \dots, w_L$, ,
19         $\hat{c}_{j,1}$, , $\dots$, $\hat{c}_{j,N}$, );
20         **for** *n=1:N* **do**
21            $w^{j,n} \leftarrow$ Replace $t_j$ in $w$ with $\hat{c}_{j,n}$;
22            Input $w^{j,n}$ into $\Phi$ to obtain the probability of correct sentiment prediction $P(\hat{y} = \tilde{y})_{j,n}$ ;
23            $\theta_{j,n} \leftarrow \beta P(\hat{y} = \tilde{y})_{j,n}^2$ ;
24            Compute $\mathcal{L}_{j,n}^{(ls)}$ by feeding $s$ with $\hat{c}_n$ labeled as true substitute into $\Psi$ ;
25            $\mathcal{L}_{j,n}^{(ls)} \leftarrow \theta_{j,n} \mathcal{L}_{j,n}^{(ls)}$;
26         **end**
27      **end**
28      $\mathcal{L}^{(ls)} \leftarrow \mathcal{L}_{1,1}^{(ls)} + \dots + \mathcal{L}_{1,N}^{(ls)} + \mathcal{L}_{J,1}^{(ls)} + \dots + \mathcal{L}_{J,N}^{(ls)}$;
29      Compute sentiment analysis loss $\mathcal{L}^{(sa)}$ using $w^f$ as input;
30      $L \leftarrow \mathcal{L}^{(sa)} + \mathcal{L}^{(ls)}$ ;
31   **end**

---

During the inference stage, we choose the candidate $\hat{c}$ whose corresponding hidden state $r_k$ is the most similar to $u_t$, measured by Euclidean distance:

$$\hat{c} = \arg\min(d(u_t, r_k)) \qquad (9)$$

We then use the resulting lexical substitution model to find replacements for the selected target words in sentiment analysis input. To determine which words are selected as disambiguation tar-

gets, we select the top $I$ words in the input with the highest $a$ produced by Equation 3, forming the set $w^{att}$. We denote their corresponding representations as $V^{att} = \{v_1^{att}, \ldots, v_I^{att}\}$. For each $v_i^{att}$, we compute its average Euclidean distance to all of its candidates' hidden states. The candidates consist of WordNet synonyms of all of its possible senses under the same part-of-speech type, which are transformed into hidden states $G = \{g_1, \ldots, g_M\}$ using Equation 2. $M$ represents the number of candidates provided by WordNet. The top $J$ words in $w^{att}$ with the largest corresponding average distance are considered to be the ones with the most diverse word meanings, and thus are more likely to require disambiguation. We will demonstrate the success of such an ambiguous word selection method in the later ablation study (see Section 6.1).

Hence, target words $t = (t_1, \ldots, t_J)$ are selected by finding each corresponding $v_j^{att}$:

$$v_j^{att} = \arg\max_i \left(\frac{1}{M} \sum_M d(v_i^{att}, g_m)\right). \quad (10)$$

Following the lexical substitution benchmarks, only words whose part-of-speech tags fall under the general categories of noun, verb, adjective, and adverb are considered as targets.

### 4.3 Dynamic Rewarding Mechanism

To fine-tune the lexical substitution model on the downstream sentiment analysis task without knowing WSD gold standard truth, for each target word $t_j$, we utilize the top $N$ candidates $\hat{c}_j = \{\hat{c}_{j,1}, \ldots, \hat{c}_{j,n}, \ldots, \hat{c}_{j,N}\}$ produced by the lexical substitution model. The new input resulting from $t_j$ being replaced by candidate $\hat{c}_{j,n}$ is denoted as $w^{j,n}$. Same with Equation 5, the probability distribution of a sentiment prediction from $w^{j,n}$ is:

$$P(\hat{y})_{j,n} = softmax(FNN_2(h^{j,n})), \quad (11)$$

where $h^{j,n}$ is the hidden states of $w^{j,n}$ outputted by Equation 4 in the sentiment analysis model.

To adjust the model in such a way that a more accurate sentiment prediction results in a higher reward for the corresponding substitution output, we compute the loss weight for $\hat{c}_{j,n}$ being the correct substitution prediction as:

$$\theta_{j,n} = \beta P(\hat{y} = \tilde{y})_{j,n}^2, \quad (12)$$

where $\beta$ is a hyperparameter for balancing the sentiment analysis and lexical substitution losses, $\tilde{y}$ is the ground truth sentiment label as defined above.

| Dataset | Split | # Samples |
|---|---|---|
| SemEval 2007 Task 10 | - | 2,010 |
| CoinCo | - | 15,629 |
| | Train | 8,530 |
| Rotten Tomatoes | Dev | 1,066 |
| | Test | 1,066 |
| Financial PhraseBank | - | 4,846 |
| STS-Gold | - | 2,026 |

Table 2: Datasets used for experiments.

We formulate the input $s^j$ to lexical substitution model with sentence $w$ and candidates $\hat{c}_j$, using Formula 1. Similar to Equation 8, the loss $(\mathcal{L}_{j,n}^{(ls)})$ of the lexical substitution model when $\hat{c}_{j,n}$ is considered as gold standard is computed as follows:

$$\mathcal{L}_{j,n}^{(ls)} = -\sum_i \log \frac{exp(d(u_t, \hat{c}_{j,n})/\tau)}{\sum_j exp(d(u_t, \hat{c}_j)/\tau)}. \quad (13)$$

Finally, the total loss is computed as:

$$\mathcal{L} = \mathcal{L}^{(sa)} + \theta_{1,1}\mathcal{L}_{1,1}^{(ls)} + \cdots + \theta_{J,N}\mathcal{L}_{J,N}^{(ls)}. \quad (14)$$

## 5 Experiments

### 5.1 Datasets

To fine-tune the pretrained lexical substitution model, we combined **SemEval 2007 Task 10** (McCarthy and Navigli, 2007) and **CoinCo** (Kremer et al., 2014). The former consists of 2,010 example sentences for 201 polysemous words. The latter contains 2,474 sentences with multiple words to be substituted. We used 70% of the concatenated dataset for training, 30% for testing.

For sentiment analysis, we used three datasets from different domains: **Rotten Tomatoes** (Pang and Lee, 2005) consists of 5,331 positive and 5,331 negative sentences collected from Rotten Tomatoes movie reviews. **Financial PhraseBank** (Malo et al., 2014) contains 4,846 sentences from English financial news, each labeled as negative, positive, or neutral. As the dataset has no standard split, We adopted a split of 70% and 30% for the training and testing sets. **STS-Gold** (Saif et al., 2013) contains 2,026 tweets, annotated as negative or positive. As the dataset has no standard split, We adopted a split of 90% and 10% for the training and testing sets. Details of the datasets are listed in Table 2.

### 5.2 Baselines

Since we aim to improve a sentiment analysis classifier with explicit WSD in a neurosymbolic fashion, vanilla **RoBERTa**-large (Liu et al., 2019) is

considered as the most direct baseline, because we also used it for sentiment classification. To put our model's performance into perspective, we further include published baselines with the highest reported accuracy on the experiment datasets, which consist of various pre-trained language models fine-tuned on the target dataset, i.e., **FinBERT** (Araci, 2019), **DeBERTa** (He et al., 2020; Sileo, 2023); and ones trained with additional resources, namely, **DeBERTa+tarksource** (Sileo, 2023) which is pre-trained on a unified dataset for 500 English tasks, and **SenticNet 7** (Cambria et al., 2022) which is a XLNet enhanced by sentiment knowledge.

### 5.3 Setups

To obtain the lexical substitution model, we fine-tune ALM with the lexical substitution datasets for 20 epochs with a batch size of 20 and a learning rate of 1e-8. The pre-trained model we used as the encoder for sentiment analysis is RoBERTa-large. The hidden size for both models is set to 1024. We use the NLTK part-of-speech tagger (Bird, 2006) to label the sentiment analysis input.

To train our proposed framework, we adopt $I = 5$, $J = 2$, $K = 15$, $N = 3$, $\tau = 0.05$, and $\beta = 0.0001$. We set the learning rate to 1-e6. The models are trained for 50 epochs with a batch size of 10, using early stopping. We used Adam (Kingma and Ba, 2014) as the optimizer. Experiments are run on RTX A6000 GPU. Following the setups in the baselines, we adopt accuracy as an evaluation metric.

## 6 Results

We compare our model with the baselines in Table 3. Our model outperforms existing models with the best-published results, achieving state-of-the-art performance on the three datasets. Compared to the RoBERTa baseline, our model is able to improve the accuracy of sentiment analysis on corpora from different domains to varying extent. We hypothesize that the extent of improvement is dependent on domain types. For instance, Twitter language is informal and brief, with frequent use of short homographs. Thus, the integrated disambiguation model is able to obtain the most significant increase of 1.91% accuracy on STS-Gold. In the financial domain, metaphors and less common word senses are commonly used, accounting for our 1.23% gain on Financial PhraseBank.

Movie reviews, on the other hand, often use

| Model | RT | FPB | STS-Gold | Avg |
|---|---|---|---|---|
| DeBERTa | 90.42 | 84.48 | - | 87.45 |
| +tasksource | 90.99 | 85.20 | - | 88.10 |
| FinBERT | - | 86.00 | - | - |
| SenticNet 7 | - | - | 90.08 | - |
| RoBERTa | 91.31 | 87.09 | 97.14 | 91.85 |
| Ours | **91.87*** | **88.32*** | **99.05*** | **93.08** |

Table 3: Comparisons with baselines. RT stands for Rotten Tomatoes. FPB stands for Financial PhraseBank. Bold font denotes the best results. The RoBERTa baseline was implemented by us. * denotes the improvement is statistically significant ($p < 0.01$ on a two-tailed t-test), against the highest baseline score. Our results are averaged over 5 runs.

standard expressions that are semantically clearer. Hence, our model obtains a marginal increase of 0.56% on Rotten Tomatoes. Despite the improvement being marginal on average, i.e., 1.23%, we would like to highlight that this improvement is significant considering the context of error propagation from the lexical substitution model. Our framework consistently demonstrates improvements across all datasets, substantiating its ability to achieve higher gains in accuracy while minimizing any potential loss. Furthermore, we can interpret the WSD outcomes and the sentiment analysis decision-making process before and after lexical substitutions from the proposed neurosymbolic framework. We will demonstrate the interpretability of the framework in Section 6.2.

### 6.1 Ablation Study

To prove the effectiveness of each component in our proposed framework, we compare with two variations: **w/o WSD**, where sentiment prediction is done by the sentiment analysis model without any modifications to the input; **w/o DR**, where the lexical substitution model is not fine-tuned during sentiment analysis training. To prove our hypothesis in WSD target word selection, we compare the proposed framework with two variations: **Top-2**, where the top 2 words in the input with the highest attention weights are automatically chosen as target words to be disambiguated; **Rand**, where 2 out of the most attended 5 words are randomly selected to be the target words. The comparison of performance is shown in Table 4. It can be observed that our full model outperforms w/o WSD by 0.28% on Rotten Tomatoes, 1.03% on Financial PhraseBank, and 1.43% on STS-Gold.

The extent of gains in different domains is in

| Model | RT | FPB | STS-Gold | Avg |
|---|---|---|---|---|
| w/o WSD | 91.59 | 87.29 | 97.62 | 92.17 |
| w/o DR | 91.40 | 87.27 | 96.67 | 91.78 |
| Top-2 | 91.60 | 87.81 | 97.62 | 92.34 |
| Rand | 89.72 | 86.30 | 97.14 | 91.05 |
| Ours | **91.87** | **88.32** | **99.05** | **93.08** |

Table 4: Ablation study results.

line with our hypothesis in the previous section. It shows that our disambiguation framework is indeed able to improve the accuracy of sentiment analysis by selecting the appropriate ambiguous words and paraphrasing them into semantically clearer ones. Our model also achieves higher accuracy than w/o DR on all three datasets, proving the validity and effectiveness of our dynamic rewarding mechanism. Noticeably, without dynamic rewarding and joint fine-tuning, the performance of w/o DR is weaker than w/o WSD, showing that dynamic rewarding-based fine-tuning with data from another task can effectively alleviate error propagation from lexical substitution.

Our proposed framework outperforms the Top-2 selection method on all datasets. It proves our assumption that words that are highly diverse in their senses are more likely to cause ambiguity for sentiment prediction. Rand performs the worst on all three datasets compared to w/o WSD, Top-2, and our model, indicating i) the necessity of selecting the appropriate target words to disambiguate so as not to introduce noise, and ii) the validity of choosing target words based on both their importance to the decision and their sense diversity level.

## 6.2 Interpretability Demonstration

Since an interpretable encoder (HAN) is used in our framework to learn sequential representations, we can understand which words contribute the most to a sentiment analysis prediction by visualizing the attention weights of HAN.

As shown in Figure 2, we conduct a study on the change of attention weights before and after paraphrasing. In Figure 2 (a), the words that receive the most attention in the original sentence are *not*, *a*, *sore*, etc., among which *not* has significantly higher weight. This indicates that the model is mostly relying on negation to make inference. As *not* does not have diverse meaning and *a* is an article, *sore* is selected as target word and replaced with *painful*. Our attention module assigns a much higher weight to the negative adjective after paraphrasing.

This shows that our model is able to find suitable

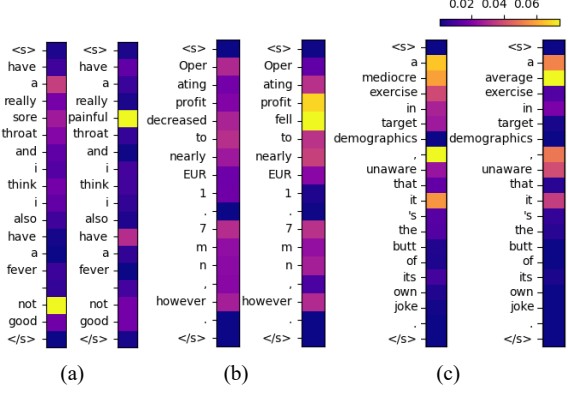

Figure 2: The most attended word demonstration before and after paraphrasing in sentiment analysis predictions. The lighter the color, the higher the attention weight.

word substitutions that aid the correct sentiment predictions. From Figure 2 (b), we can observe a rather evenly distributed attention weights in the original sentence, meaning that the attention module cannot identify the words that significantly affect sentiment. After our framework replaces the word *decreased* to *fell*, the model is able to assign higher weights to the sentiment-loaded phrase *profit fell*. From Figure 2 (c), we can see that after replacing the word *mediocre* with *average*, the sentiment related words including the paraphrased word and *unaware* all received higher attention. By attending to the straightforward paraphrases expressing opinions, the sentiment analysis model can yield correct predictions for all three cases.

## 6.3 Case Study

In Table 5, we study the ability of our framework to identify and paraphrase ambiguous words into unequivocal ones. In the first instance, albeit not affecting the prediction, the word *delicious* is disambiguated by our framework into its figurative meaning *delighted* instead of its literal one. In the second instance, the word *succeeds* has a more positive word sense equating to *win*, and a more neutral one equating to *follow*. Our framework is able to paraphrase it into the latter. In the last instance, the word *visibility* falls under the word sense of *degree of exposure to public notice* in this context. Our framework correctly paraphrases it into *profile*, which is the synonym under the same sense in WordNet. It can be concluded that our framework can disambiguate word senses in a transparent and interpretable way for sentiment analysis.

| Setup | Content |
|---|---|
| Original | a delicious and delicately funny look at the residents of a copenhagen neighborhood coping with the befuddling complications life tosses at them. |
| Paraphrased | a delighted and delicately funny look at the residents of a copenhagen neighborhood coping with the befuddling complications life tosses at them. |
| Original | Le Lay succeeds Walter G++nter and will be basedin Finland. |
| Paraphrased | Le Lay follows Walter G++nter and will be based in Finland. |
| Original | Strong brand visibility nationally and regionally is of primary importance in home sales, vehicle and consumer advertising. |
| Paraphrased | Strong brand profile nationally and regionally is of primary importance in home sales, vehicle and consumer advertising. |

Table 5: Case study of disambiguation as paraphrasing.

| Setup | Content | Label |
|---|---|---|
| Original | a tour de force of modern cinema. | Positive |
| Paraphrased | a tour de power of modern cinema. | Positive |
| Original | Is just at work for the day -sigh- I have a headache | Negative |
| Paraphrased | Is good at work for the day -sigh- I have a headache | Positive |
| Original | the skills of a calculus major at m.i.t. are required to balance all the formulaic equations in the long-winded heist comedy who is cletis tout? | Negative |
| Paraphrased | the science of a calculus major at m.i.t. are required to balance all the formulaic equations in the long-winded heist comedy who is cletis tout? | Positive |

Table 6: Error analysis. Red indicates errors made by our proposed framework.

## 6.4 Error Analysis

Finally, we perform a brief error analysis (Table 6). In the first sentence, the model wrongly paraphrases *force* into *power*, because it does not recognize the multi-word idiom *tour de force*. It shows the model's limitation in understanding the semantic meaning of infrequent multi-word idiom. In the second sentence, the word *just* is wrongly paraphrased as *good*, which negatively influenced the sentiment prediction. This is mainly due to the part-of-speech tagger wrongly classifying the word *just* into a adjective, instead of an adverb. Thus, the lexical substitution module might be subjected to error propagation of the part-of-speech tagger and introduce noise into the input. In the last sentence, the word *skills* is paraphrased as *science*. It is likely that the model mistakes the word *major* as a field of study instead of a student studying a field. Thus, the substitution makes sense in the local context *the science of calculus major*, but not in the whole context, meaning that the model fails to understand the semantic meaning of the sentence.

## 7 Conclusion

In this work, we propose a neurosymbolic framework for sentiment analysis that incorporates the disambiguation of word senses by identifying and paraphrasing ambiguous words in the input, which is realized by leveraging a pre-trained lexical substitution model. The proposed framework not only improves the accuracy of sentiment analysis, but also achieves interpretability by enabling us to trace back which words are paraphrased and what words they are replaced with.

Additionally, to fine-tune the lexical substitution model to provide better disambiguating substitutes for sentiment analysis with no ground-truth word sense labels, we adopt the dynamic rewarding mechanism to jointly train the sentiment analysis and lexical substitution models. Experiments show that our framework is indeed effective in improving the performance of sentiment analysis, obtaining state-of-the-art performance on three corpora from different domains.

## Limitations

The framework proposed in this paper have to be seen in light of some limitations. First, the performance of sentiment analysis to some extent depends on a decent-performing pre-trained lexical substitution model. Currently, polysemous samples in lexical substitution data are not as abundant as in WSD data, which might limit its ability in disambiguating less common polysemies. Second, the effectiveness of our proposed model is only tested on English corpora, as it relies on the sense knowledge from WordNet. The disambiguation target selection and paraphrasing methods might work on languages with more complex segmentation or limited sense lexicon.

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
