# OpenReview forum: "Neuro-Symbolic Sentiment Analysis with Dynamic Word Sense Disambiguation"
_EMNLP/2023/Conference — EMNLP 2023 Findings_

### Official Review · Reviewer_PBHS · 2023-07-26

**Soundness:** 3

**Excitement:**

2: Mediocre: This paper makes marginal contributions (vs non-contemporaneous work), so I would rather not see it in the conference.

**Missing References:**

- Related work: Fine-Grained Contextual Predictions for Hard Sentiment Words, Ebert and Schuetze, EMNLP 2014

**Paper Topic And Main Contributions:**

The paper describes a multi-step approach for Word Sense Disambiguation (WSD) specifically for Sentiment Analysis (SA) (although the approach is task agnostic). In a first step, words that have high importance for the SA task are identified using an attention mechanism. For the most important words they identify synonyms via WordNet and rank them via a dedicated "lexical substitution model". The highest ranking candidate then substitutes the original target word and the so modified input sentence is used as input for a sentiment classifier.

**Questions For The Authors:**

- QA: What's the "dynamic rewarding mechanism"?
- QB: Alg. 1:
	- I'm very confused by L15-27. There seem to be some discrepancies to the text in Sec. 4. I may be misunderstanding a lot here, so please take the following with a grain of salt. Here's what I think should be corrected:
	- L16: According to Sec. 4.2, $\psi$ isn't supposed to "produce top N candidates$ but rather representations if $w^{j,n}$ and $c^hat_{j,n}$.
	- $s$ in L18 is never used.
	- L21: According to Sec. 4.3 $w^s$ should be $w^{j,n}$ and $c^hat_n$ should be $c^hat_{j,n}$.
- QC: Looking at the hyperparameters in Sec. 5.3 I assume you ran some kind of hyperparameter search. How was that done? Given that the Financial PhraseBank and STS-Gold dataset don't have a dev set it would be problematic to tune on the test set.
- QD: L450: Reported results are not comparable to previous work, if the splits are different. That being said, previous work on Financial PhraseBank shows that SotA performance is far above 90% (https://paperswithcode.com/sota/text-classification-on-financial-phrasebank). This in itself is not a reason to reject the paper, but the statement should be toned down to reflect current state of the art. I further suggest adding the actual SotA results for all datasets into the table.
- QE: Sec. 6.1.
	- What's the difference between the "w/o WSD" model and the RoBERTa baseline?
	- A reasonable baseline would be to use a standard WSD system instead of the substitution model.

**Reasons To Accept:**

The presented approach allows to interpret the importance of words w.r.t. polarity and their replacement candidate.

**Reasons To Reject:**

- The presented approach is complex and requires 2 models and a lexical resource.
- Generally, I had trouble understanding Sec. 4 and the algorithm. Unfortunately, Fig. 1 didn't help in shedding more light to the algorithm.
- I had trouble following the notation throughout the paper. A table with all used symbols would have helped, because the variable definitions are spread over multiple pages. Some of the notation is inconsistent (more details below).
- It's difficult to judge how well the presented system works given that only 1 out of 3 of the datasets has a published train/dev/test split where numbers can be compared to previous research. On that dataset the improvement is marginal (but still significant).

**Reproducibility:**

2: Would be hard pressed to reproduce the results. The contribution depends on data that are simply not available outside the author's institution or consortium; not enough details are provided.

**Reviewer Confidence:**

3: Pretty sure, but there's a chance I missed something. Although I have a good feel for this area in general, I did not carefully check the paper's details, e.g., the math, experimental design, or novelty.

**Typos Grammar Style And Presentation Improvements:**

- Fig. 1: "Lexical Substitution" has the "n" in an extra line.
- There seems to be some inconsistency in the use of $K$ and $N$. L210, L316 use $K$ as number of candidates, but Fig. 1 and L255 use $N$.
- L413: typo in "neurosymbolic"
- Missing a reference to Tab. 1 to Sec. 5.1.

---

> ### Author Rebuttal · Authors · 2023-08-28
>
> Thank you for your recognition of the novelty and contributions of our work. We hope our work can provide a new perspective of disambiguating word senses for downstream tasks that is effective and explainable.
>
> QA: As we describe in Section 4.3, since we do not have gold standard truth annotation for which substitution candidate is the most appropriate, we compute the loss of each candidate being the “truth” (best substitute) respectively, but weight each loss with the probability of correct sentiment prediction when the original target word is replaced by the corresponding candidate. In other words, if the substitution of a candidate word makes the sentiment analysis model more likely to make the correct prediction, we reward the lexical substitution model for ranking it higher by increasing the corresponding loss weight. This mechanism is used to further finetune the lexical substitution model so that it can adjust its output according to the downstream task and the text domain.
>
> QB: (1) “L16”. Thank you for pointing out our careless mistake! We attach the correct version of L15-27 for preview. Since in Alg 1 all candidates $\hat{c}$ is only used locally in the $j=1:J$ loop (L15-27), we omitted the subscript $j$ for a more elegant algorithm presentation in the original script, but we can see how different expression from the main text might rise confusion. We will modify the $j=1:J$ loop so that all notation matched with the text, as shown below. (2) “s in L18” is used in L24, which correspond to the $s^j$ Line 380-382 in text.
>
>     For $j=1:J$
>         Input $w$ into $\Psi$ to produce top $N$ candidates $\hat{c}_j=(\hat{c}_{j,1},\dots,\hat{c}_{j,N})$ for target word $t_j$, ordered by probability;
>         Replace $t_j$ in $w^f$ with $\hat{c}_{j,1}$;
>         $s \leftarrow (\textrm{<s>},w_1,\dots,t_i,\dots,w_L,\textrm{</s>}, \hat{c}_{j,1},\textrm{</s>},\dots, \hat{c}_{j,N},\textrm{</s>})$;
>         For $n=1:N$
>                 $w^{j,n} \leftarrow$ Replace $t_j$ in $w$ with $\hat{c}_{j,n}$;
>                 Input $w^{j,n}$ into $\Phi$ to obtain the probability of correct sentiment prediction $P(\hat{y}=\tilde{y})_{j,n}$ ;
>                 $\theta_{j,n} \leftarrow \beta P(\hat{y}=\tilde{y})_{j,n}^2$ ;
>                 Compute $\mathcal{L}^{(ls)}_{j,n}$ by feeding $s$ with $\hat{c}_n$ labeled as true substitute into $\Psi$ ;
>                 $\mathcal{L}^{(ls)}_{j,n} \leftarrow \theta_{j,n}\mathcal{L}^{(ls)}_{j,n}$\;
>
> QC: We follow the setting of $\tau$ in Mao et al. (2023) and learning rate of the best published baselines. $\beta$ is fixed across experiment on all datasets.
>
> QD: Thank you for your suggestion. We did not consider unpublished results when including baselines. We will update the SOTA.
>
> QE: (a) “w/o WSD” is Roberta with the addition of two HAN blocks (see the sentiment analysis component in Fig. 1).
>
> (b) Standard WSD systems face the following gaps to be integrated into sentiment analysis: 1) target words in the input must be predefined, 2) the direct output of current SOTA WSD systems is the gloss definition of a target word (Song et al., 2021; Barba et al., 2021a; Barba et al., 2021b), which is difficult to embed into sentiment analysis model in an explainable way, and 3) there is no straightforward way to finetune a standard WSD system to the target domain due to lack of sense annotations in downstream task corpora. In contrast, our framework consists of interdependent components that aim to bridge these gaps.

---

### Official Review · Reviewer_PXHm · 2023-08-01

**Soundness:** 3

**Excitement:**

3: Ambivalent: It has merits (e.g., it reports state-of-the-art results, the idea is nice), but there are key weaknesses (e.g., it describes incremental work), and it can significantly benefit from another round of revision. However, I won't object to accepting it if my co-reviewers champion it.

**Paper Topic And Main Contributions:**

This paper proposes a novel framework for sentiment classification, which leverages word sense disambiguation (WSD) as an auxiliary lexical substitution task. The proposed framework can pick up the important and polysemous words for sentiment classification, and replace them with semantically unequivocal words. To do this, authors develop an interpretable attention module for word selection and introduce a pre-trained language model ALM for lexical substitution. Additionally, a self-supervised dynamic rewarding mechanism is designed to further fine-tune ALM. The experimental results show that the proposed framework is more accurate and interpretable.

**Questions For The Authors:**

1) It would be better to analyze the time efficiency of the preposed framework.
2) In Line 16 Algorithm 1, does only the word t_j is feed into pre-trained lexical substitution model?
3) In my opinion, after word selection, fusion information of synonyms via attention mechanisms is a simple way to use WordNet. Could this method be used as a baseline?
4) There are so many symbols in this paper that it is not very readable.


**Reasons To Accept:**

1) The disambiguation of polysemous words in sentiment analysis remains under-explored. The proposed framework is a reasonable way to boost sentiment classification with WSD.
2) With the lexical substitution, the framework is more explainable and interpretable.


**Reasons To Reject:**

1) The proposed framework is very time-consuming during both the training and inference phases.
2) The experimental results are not very convincing. None of the chosen baselines in Table 2 uses the extra resource WordNet. Additionally, the strongest baseline is a vanilla RoBERTa model.


**Reproducibility:**

3: Could reproduce the results with some difficulty. The settings of parameters are underspecified or subjectively determined; the training/evaluation data are not widely available.

**Reviewer Confidence:**

4: Quite sure. I tried to check the important points carefully. It's unlikely, though conceivable, that I missed something that should affect my ratings.

---

> ### Author Rebuttal · Authors · 2023-08-28
>
> Thank you for your recognition of the novelty and contributions of our work. We hope our work can provide a new perspective of disambiguating word senses for downstream tasks that is effective and explainable.
>
> “time-consuming”. It's a trade-off between gaining explainability with a neuro-symbolic method and a pure neural network-based method during the training stage. To mitigate error propagation of lexical substitutions, we have to take an extra step to train the model with multi-lexical substitutions and dynamic rewarding to learn the most robust one. This is because there is no parallel dataset with word sense and sentiment labels that allows us to learn both tasks in a multitask learning fashion. However, during the inference stage, the output can be achieved with a single feedforward propagation. Additionally, the inference phrase is actually quite efficient, because the lexical substitution model no longer needs computational time to be finetuned, and the time needed to replace target words in an input sentence is quite marginal, compared to other related works (Pu et al., 2018; Campolungo et al., 2022) that use a WSD system to disambiguate every word in the sentence.
>
> “Line 16 Algorithm 1”. Thank you for pointing out our careless mistake! The correct step would be: “Input $w$ into $\Psi$ with $t_j$ as target to produce top $N$ candidates $\hat{c} = (\hat{c}_1, \dots, \hat{c}_N)$”. We further make modification of L15-27 so that the notations in Alg 1 will be exact match of the notations in text.
>
>     For $j=1:J$
>         Input $w$ into $\Psi$ to produce top $N$ candidates $\hat{c}_j=(\hat{c}_{j,1},\dots,\hat{c}_{j,N})$ for target word $t_j$, ordered by probability;
>         Replace $t_j$ in $w^f$ with $\hat{c}_{j,1}$;
>         $s \leftarrow (\textrm{<s>},w_1,\dots,t_i,\dots,w_L,\textrm{</s>}, \hat{c}_{j,1},\textrm{</s>},\dots, \hat{c}_{j,N},\textrm{</s>})$;
>         For $n=1:N$
>                 $w^{j,n} \leftarrow$ Replace $t_j$ in $w$ with $\hat{c}_{j,n}$;
>                 Input $w^{j,n}$ into $\Phi$ to obtain the probability of correct sentiment prediction $P(\hat{y}=\tilde{y})_{j,n}$ ;
>                 $\theta_{j,n} \leftarrow \beta P(\hat{y}=\tilde{y})_{j,n}^2$ ;
>                 Compute $\mathcal{L}^{(ls)}_{j,n}$ by feeding $s$ with $\hat{c}_n$ labeled as true substitute into $\Psi$ ;
>                 $\mathcal{L}^{(ls)}_{j,n} \leftarrow \theta_{j,n}\mathcal{L}^{(ls)}_{j,n}$\;
>
> “baseline”. As stated in Line 412-413, the aim of our experiment is to prove that our framework of incorporating WSD, specifically lexical substitution as explainable explicit WSD, can improve the performance of a sentiment analysis model. We include other baselines mainly to put our model’s performance into perspective. (Please note that DeBERTa+tarksource uses a unified dataset containing much more training samples than ours.) Since WordNet is essentially a word sense dictionary, it is difficult to incorporate it without some measure to disambiguate senses, which is the problem our paper is trying to address. If my understanding of the baseline suggestion is correct, fusing synonyms of the target word under all of its sense, albeit via attention mechanism, would incorporate more “noisy” word senses into the input context, which does not quite make sense regarding the disambiguation goal.

---

### Official Review · Reviewer_xJze · 2023-08-03

**Soundness:** 3

**Excitement:**

3: Ambivalent: It has merits (e.g., it reports state-of-the-art results, the idea is nice), but there are key weaknesses (e.g., it describes incremental work), and it can significantly benefit from another round of revision. However, I won't object to accepting it if my co-reviewers champion it.

**Paper Topic And Main Contributions:**

Summary: The paper presents an approach to incorporate word sense disambiguation into sentiment analysis. WSD is modeled as a word substitution task. The claim of the paper is that by substituting the high-attention yet ambiguous words can make the sentence easier as an input for sentiment analysis. The authors report results on fine-tuned RoBERTa, FinBERT etc. The paper also presents a qualitative analysis of sentences where words are correctly and incorrectly substituted.

Contribution: Using substitution of words for a NLP task is an interesting recent idea. It has been shown to work for detection of AI-generated text (to an extent; in the DetectGPT paper).

**Questions For The Authors:**

1) Is the reliance on WordNet a limitation of the work since WordNet may not potentially capture all sense of a word?

2) The proposed approach replaces one word at a time. However, WSD is known to be a task where the meaning of a word is influenced by the context. How does the proposed model take into account the importance of context in WSD?

3) The idea of using WSD in conjunction with sentiment analysis is not new:
(a) Rentoumi, Vassiliki, George Giannakopoulos, Vangelis Karkaletsis, and George A. Vouros. "Sentiment analysis of figurative language using a word sense disambiguation approach." In Proceedings of the International Conference RANLP-2009, pp. 370-375. 2009.
(b) https://aclanthology.org/W15-2916.pdf
How does the current work

4) How does the model take into account the requirement that the substituted word must also be a high-attention word for the classification task?

5) What is neuro-symbolic about the approach - that allows it to be distinguished from, say, multi-task learning? It may be useful to highlight that using prevalent paradigms for neurosymbolic methods.

6) Related to 5a: The idea of using a model for a certain task in order to serve as input (or vectors) for another task is not new. For example: https://arxiv.org/abs/1906.05466 . How does this model improve upon such approaches in being neuro-symbolic?

7) What is the intuition behind using Euclidean distance to choose candidate substitutions?

**Reasons To Accept:**

1) The paper models WSD as a word substitution task - and shows that doing so can improve the performance of sentiment analysis.

**Reasons To Reject:**

1) It is not clear what makes the approach 'neuro-symbolic'. Please refer to questions 3, 5,6 below.

2) The work relies on a static resource such as WordNet. Incompleteness of WordNet may be a limitation of the work.

3) The idea of modeling WSD as a substitution is interesting. However, WSD must take into account other words in the context of the sentence. It is not clear how and if the model does this.

4) The baselines seem rather weak in comparison to the proposed approach. The ablation study offers comparable models - and shows promise. However, other methods where WSD is used to aid sentiment analysis would have made for a stronger comparison.

**Reproducibility:**

3: Could reproduce the results with some difficulty. The settings of parameters are underspecified or subjectively determined; the training/evaluation data are not widely available.

**Reviewer Confidence:**

2: Willing to defend my evaluation, but it is fairly likely that I missed some details, didn't understand some central points, or can't be sure about the novelty of the work.

**Typos Grammar Style And Presentation Improvements:**

Figure 1 is quite loaded with labels on the edges being explained in the description. I would suggest to simplify Figure 1 and extend the details to a new figure.


Line 212: Is there a special token to indicate which word was replaced? The equation does not seem to say so.

Eulidean -> Euclidean (in the algorithm)

It may be useful to add a section on neuro-symbolic methods in the related work section.

---

> ### Author Rebuttal · Authors · 2023-08-28
>
> Thank you for your recognition of the novelty and effectiveness of our work. We hope our work can provide a new perspective of disambiguating word senses for downstream tasks that is effective and explainable.
>
> Q1. The reliance on WordNet is a limitation of the WSD task as a whole. To the extent of our knowledge, supervised WSD was commonly trained and tested on datasets annotated using static WordNet sense definitions, while most unsupervised WSD also relied on existing knowledge bases [1, 2]. The generation of word senses is a challenge in WSD that is outside the research scope of this paper.
>
> Q2. It is worth noting that, in most WSD systems (Blevins and Zettlemoyer, 2020; Bevilacqua and Navigli, 2020; Song et al., 2021; Barba et al., 2021a), a target word in a sentence is disambiguated individually without considering the word sense of other target words in the sentence, which is essentially the same as our setting. Therefore, (1) our model learns the semantic context of the input sentence implicitly via pretrained language model, same as in other WSD works, (2) replacing of a target word independently from other target words has the advantage of minimizing error propagation, where a wrong substitute negatively affects the substitution of the other target word.
>
> Q3. We appreciate the suggestion of related works. The question is incomplete, so this answer is based on our assumption on what the question is. In work (a), the polarity of each word is assigned on sense-level. However, a word with neutral sense might have a positive or negative meaning in the context of the sentence, which our model would be able to discern. Work (b) has the same limitation as the works we discussed in Line 170-183.
>
> Q4. The model does not require the substituted word to be high-attention. Our aim is to replace important words with diverse sense so that the input would be less ambiguous to the model. The high attention score of substituted word shown in Fig.2 is the outcome of the disambiguated input, but not the goal of our model. There are also instances where the high-attention words in the original sentence likely should not receive a high attention score due to its prevailing sense, e.g., the second example sentence in Table 4.
>
> Q5&6. Our model is neuro-symbolic in the sense that the word senses are explicitly disambiguated for the downstream task with other symbolic representations, which contrasts with the exclusive reliance on neural networks [3]. By definition, the work mentioned in Q6 can be considered neuro-symbolic. The aim of our paper is to propose a new paradigm where the disambiguation of word sense can be integrated into sentiment analysis (and other downstream classification task) in an intuitive and explainable way. We appreciate the suggestions but such comparisons are very much outside of our research scope.
>
> Q7. As stated in Line 301-302, to choose the appropriate substitution, we utilize ALM and adopt  similar training process from Mao et al. (2023), where using Euclidean distance as the similarity measure performs the best.
>
> “Line 212”. We do not implement a special token to indicate which word is replaced. We use an index sequence to keep track of the position of target words, same as in Mao et al. (2023).
>
> [1] Bevilacqua, Michele, et al. "Recent trends in word sense disambiguation: A survey." Proceedings of the Thirtieth International Joint Conference on Artificial Intelligence, IJCAI-21. International Joint Conference on Artificial Intelligence, Inc, 2021.
>
> [2] Maru, Marco, et al. "Nibbling at the hard core of Word Sense Disambiguation." Proceedings of the 60th Annual Meeting of the Association for Computational Linguistics (Volume 1: Long Papers). 2022.
>
> [3] Cambria, Erik, et al. "Sentiment analysis is a big suitcase." IEEE Intelligent Systems 32.6 (2017): 74-80.

---

### Meta-Review · Area_Chair_tRsu · 2023-09-15

**Recommendation:** 3

**Metareview:**

The paper presents a novel neuro-symbolic approach that models WSD as a word substitution task to improve sentiment analysis performance. It identifies the ambiguous words and substitutes them using candidates from WordNet.
The approach is considered reasonable by the reviewers. It offers interpretability advantages and improves the sentiment analysis performance. However, the reviewers raised concerns regarding the reliance on static resources like WordNet, limited baselines, the time-consuming nature of this approach, and the experimental results. The authors addressed some of these concerns in their response by providing further clarification, however, the reviewers are still quite ambivalent about the excitement of this work.

---

### Decision · Program_Chairs · 2023-10-07

**Decision:**

Accept-Findings

**Comment:**

The paper presents a novel neuro-symbolic approach that models WSD as a word substitution task to improve sentiment analysis performance. It identifies the ambiguous words and substitutes them using candidates from WordNet.
The approach is considered reasonable by the reviewers. It offers interpretability advantages and improves the sentiment analysis performance. However, the reviewers raised concerns regarding the reliance on static resources like WordNet, limited baselines, the time-consuming nature of this approach, and the experimental results. The authors addressed some of these concerns in their response by providing further clarification, however, the reviewers are still quite ambivalent about the excitement of this work.